# Small Molecules Targeting Viral RNA

**DOI:** 10.3390/ijms241713500

**Published:** 2023-08-31

**Authors:** Gregory Mathez, Valeria Cagno

**Affiliations:** Institute of Microbiology, University Hospital of Lausanne, University of Lausanne, 1011 Lausanne, Switzerland

**Keywords:** antiviral, RNA, SARS-CoV-2, IRES, programmed ribosomal frameshift

## Abstract

The majority of antivirals available target viral proteins; however, RNA is emerging as a new and promising antiviral target due to the presence of highly structured RNA in viral genomes fundamental for their replication cycle. Here, we discuss methods for the identification of RNA-targeting compounds, starting from the determination of RNA structures either from purified RNA or in living cells, followed by in silico screening on RNA and phenotypic assays to evaluate viral inhibition. Moreover, we review the small molecules known to target the programmed ribosomal frameshifting element of SARS-CoV-2, the internal ribosomal entry site of different viruses, and RNA elements of HIV.

## 1. Introduction

We lack antivirals for most viruses. The limitations in antiviral discovery are multiple. It is fundamental to find highly specific targets in order to avoid toxic effects on the host cell since viruses are intracellular parasites. Many viruses cause acute infections and, therefore, it is difficult to start the treatment at the right moment, as evidenced, for instance, by the limit of influenza therapies which have to be given within 48 h from symptom onset [1]. Furthermore, the rapid insurgence of mutations leads to resistance to antivirals, as shown by the failure of several monoclonal antibodies to neutralize the emerging variants of Severe Acute Respiratory Syndrome Coronavirus 2 (SARS-CoV-2) [2].

One possibility for overcoming these limitations is to change the approach and the target for antiviral molecules. So far, all the antivirals in commerce are targeting either viral proteins or, in rare cases, cellular proteins [3]. However, the vast majority of proteins are undruggable [4] and RNA is emerging as a promising new target due to the conserved conformations it can adopt. Despite the nontraditional target, the feasibility of the approach is demonstrated by the presence of an FDA-approved molecule acting on a human RNA splicing site, risdiplam, approved for treating spinal muscular atrophy [5].

RNA viruses can be targeted as well since they have multiple conserved secondary and tertiary structures in their genome to express superposed open reading frames (ORF) by inducing programmed ribosomal frameshifting (PRF) [6], as signals for packaging, splicing, or for the start of translation like the internal ribosomal entry sites (IRES) [7]. Importantly, in infected cells, viral RNA is often highly expressed while host RNA synthesis is shut down [8], decreasing the risk of side effects since targeted RNA will be probably more expressed than specific cellular off-target RNAs. Furthermore, a priori selection can be done to exclude molecules binding to highly expressed host RNA [9]. However, selectivity remains a challenge but the success of antivirals from in vitro to in vivo against different viral pathogens shows the feasibility of the approach [6,7,10,11,12].

An important point to carefully monitor with the increase of RNA-targeting molecules will be the development of resistance. Despite the selection with some RNA-targeting antivirals targeting the IRES showed the possible insurgence of resistance [7,13], with others targeting the PRF, it was not possible to select resistant mutants and several artificial mutations were needed to observe a loss of efficacy [11,12]. Specific RNA targets might be, therefore, more resistant to mutations, particularly in coding regions since mutations in the ORF will result not only in a change of secondary or tertiary structure but also in a possible detrimental mutation in the encoded protein, with a consequent higher barrier to resistance. However, if viruses could become resistant to RNA-targeting antivirals, the availability of molecules with multiple targets will allow combinatorial therapies to overcome the problem.

In the last years, the growing interest in RNA as an antiviral target is partly linked to the development of techniques allowing for a better understanding of the structure of RNA in living cells, together with traditional techniques to reveal a purified RNA structure in vitro (reviewed in [14]). Since we now better understand how the viral RNA is structured and what are the interactors, we have numerous possible new targets and ways to identify molecules interacting with these structures on viral RNA.

## 2. How to Discover RNA-Targeting Antivirals

### 2.1. Understanding RNA Folding

Different techniques have been optimized to assess RNA folding in vitro and in living cells. The 3D structure of viral RNA can be determined by crystallography [15], nuclear magnetic resonance (NMR) [16], or cryo-electron microscopy (cryo-EM) [6]. However, for all techniques, the limitations are linked to the flexibility of RNA allowing to solve only structures of small portions of RNA. The viral RNA is, therefore, purified or, more often, a small portion of the RNA of interest is transcribed in vitro. Examples of viral RNA structures largely studied and determined with multiple techniques are the transactivation response element (TAR) [17,18] and the REV responsive element (RRE) [19,20,21] of HIV in association, respectively, with the TAT and REV proteins, and the pseudoknot mediating the −1 PRF of SARS-CoV-2 either alone [6,22,23,24] or in association with the ribosome [6]. For the IRES, due to its length, only single domains have been characterized by crystallography or NMR [25,26,27], while, for instance, the full-length structure of HCV IRES was only predicted by the assembly of single components on low-resolution structures obtained by small angle X-ray scattering [28] or cryo-EM in complex with the ribosome [29].

In addition to the limitations of the previously described techniques, we need to consider that the viral RNA in the cell is associated with multiple proteins and in full length, which might result in a different folding. For this reason, it is important to investigate the structure of the RNA in the natural context of the infection and one of the possible ways is to probe the viral RNA with chemicals able to selectively modify nucleotides in loops and bulges but not paired ones. After adapted sequencing, it is then possible to predict secondary and tertiary structures. Different chemical probes can be used; one of the oldest is dimethyl sulfite (DMS), while more recently new probes were identified to perform selective 2′-hydroxyl acylation analyzed by primer extension and mutational profiling (SHAPE-MaP) [30]. More details on these techniques have been reviewed elsewhere [14,31].

Thanks to DMS-MaP and SHAPE-MaP, entire viral genomes in infected cells have been studied. Several works have focused on SARS-CoV-2 [32,33,34], some on flaviviruses (in particular Dengue virus [35] and Zika virus [35,36]), one on Chikungunya [37], and one on Human Immunodeficiency Virus (HIV) [38]. The viruses listed so far have as a common feature their positive sense genome, which simplifies the analysis. However, a study on the mRNA of Influenza virus [39], a negative sense RNA virus, has been done as well.

In many of these works, the identification of structures is just described or investigated by deleting the entire secondary structure identified and seeing the effect on the viral life cycle. To date, we are missing a comprehensive evaluation, from the identification of the secondary structure to finding small molecules targeting it. To do so, druggable pockets need to be selected, an in silico screening performed, and, subsequently, the compounds tested in biological assays. Otherwise, the sequence forming the secondary structure can be cloned upstream of a reporter gene and a phenotypic screen can be directly performed. The steps to perform these experiments are described in the following paragraphs.

### 2.2. In Silico Screening

In silico screening allows for the reduction of the time and cost of in vitro assays for the identification of antivirals. The starting point is the choice of the RNA model. In the Protein Data Bank, X-ray, electron microscopy, or even solution NMR structures are present. However, only 280 structures of viral RNA are available (compared to 33,137 for viral proteins), at the time of writing. They might have a high resolution but these structures were generated from crystal, frozen, or in-buffer samples. X-ray or cryo-EM structures could then be refined through classical molecular dynamics, as done by our group [12]. However, a limitation of molecular dynamics is that it does not take into consideration the interaction with accessory proteins nor the presence of other molecules in the environment. Therefore, the resulting model might not be accurate on the real folding occurring during infection. The 3D structures based on DMS-MaP or SHAPE-MaP data should therefore be privileged since they represent predictions of real intracellular structures.

In case of the appearance of a new virus or a lack of a 3D structure available, the model can be created from the sequence of the virus. Park et al. have built directly their model using the sequence of SARS-CoV [40]. PSEUDOVIEWER was used to construct the secondary structure of a SARS-CoV pseudoknot, followed by 3D visualization with Sybyl, and then minimized with molecular dynamics. As AlphaFold for proteins, machine-learning tools could be developed to predict RNA structures, such as Atomic Rotationally Equivariant Scorer (ARES) [41]. Alternative software to build 3D models from sequence is reviewed elsewhere [14,42].

In the past years, in silico software was developed and largely used for proteins. Recently, since RNA became an interesting target, some existing tools were modified or validated (such as Glide, AutoDock Vina, and DOCK6), and new tools were developed for RNA (such as MORDOR, rDOCK, and RiboDock) [43]. In the process of the creation of docking software, each tool is trained and/or validated on different structures. This bias implies that docking software might not perform well for all the different types of pockets. Therefore, validation of the docking software with a known ligand–RNA structure or the use of several docking software should be performed before the virtual screening. Unfortunately, there are still several limitations with RNA (reviewed in [43,44]). The main issue is its flexibility. For proteins, this has been neglected or limited to the region of binding. However, the flexibility of RNA needs to be considered from the beginning, before performing in silico screening. Several methods can be used: flexible (induced-fit) docking where only the atoms close to the candidate drug can move to make space to be able to accommodate the ligand. Alternatively, different conformations of the target RNA can be generated by molecular dynamics and then used as individual receptors to perform virtual screening, as done with HIV [45]. Or, after initial docking, the ligand–RNA structure can be minimized [46]. This method allows the entire system to move considering forces that are occurring on the RNA, the ligand, and their interaction. We can even go further with this technique using classical molecular dynamics. Since the system will be placed in a box filled with ions and water, it will take into consideration these additional molecules that interact with RNA and, therefore, can change the interaction map of the ligand. The binding pose can then be refined in an aqueous and neutral environment. Additionally, this simulation will give to the investigator information about the stability of the interactions. Unfortunately, this method needs more computational resources and, as stated above, accessory proteins and other compositions of ions might have a different impact on the ligand pose due to different folding of the RNA.

After obtaining the binding poses, they can be rescored. Several scoring functions are known for proteins and, nowadays, also for RNA. Unfortunately, scoring functions are algorithms ranking binding poses based on different criteria, such as physic based (such as iMDL Score, RLDOCK, MORDOR), knowledge based (such as DrugScore^RNA^, LigandRNA), or even machine-learning based (such as RNAPosers, RNAmigos) [44]. To overcome this bias, the use of multiple scoring functions is recommended, as is done with our virtual screening on SARS-CoV-2 pseudoknot [12], helping to reduce false positive hits.

Another critical point in the virtual screening against viral RNA is the selection of the pocket. In RNA, pockets were considered too polar and solvent exposed, in opposition to the pocket present in proteins. However, a study showed that the properties of RNA pockets are similar to those of proteins using all ligand–RNA structures available [47].

To avoid off-target effects, the antiviral should target a specific pocket. To increase the selectivity of the compound, screening should be done on complex structures [48]. The viral RNA structures targeted should be not common in human cells to avoid the candidate drug binding to host RNA. Among them, pseudoknots, or IRES of several viruses are complex viral structures of choice, as described in the following paragraphs. However, new complex viral RNA structures could be identified by overcoming the limits of the structural techniques. Additionally, investigators could gain selectivity with molecules that interact with distant nucleotides and not intercalate between two base pairs or bind in the major or minor groove of RNA, avoiding them binding to cellular RNA [10].

Once the hits of the screening have been selected, a filtering step to exclude molecules binding to host RNA can be performed, thanks to the presence of tools and databases of predicted human RNA structures [9,49,50]. Only molecules with weak interactions or instability in the nonspecific pockets can be retained.

After the in silico screening, it is important to validate the results with phenotypic assays, such as dual luciferase assays and, more importantly, to verify the inhibition of the compound on wild-type virus to truly validate the screening results.

### 2.3. Phenotypic Assays: Dual Luciferase Assays

Even without understanding the exact folding of a viral RNA structure, it is possible to find inhibitors with phenotypic assays. Chemical assays in which the affinity of a small molecule for a structure of interest, such as fluorescent dye displacement assays, can be used [51]. However, this type of assay does not directly allow the measurement of the biological activity of a small molecule (i.e., a molecule can bind to the target RNA without affecting its function) [52]. To couple the binding and the alteration of the biological function, the most common method is to evaluate the expression of a reporter gene under the control of the target RNA. For instance, the sequence of interest can be cloned upstream of a luciferase or in between two different sequences encoding luciferases (or fluorescent proteins) [53] in order to evaluate the expression of the reporter with high throughput screening of inhibitors [7].

The alteration of the expression of the reporter can be evaluated directly upon transfection of plasmids [11,12], through transfection of in vitro transcribed RNA, or with cell-free systems [54].

All the alternatives have advantages and disadvantages. For instance, the transfection of plasmids in cells is linked to nuclear expression of the transcript with possible mistakes or unwanted splicing events, especially for long constructs, resulting in altered expression of the reporter independent of the action of the compounds. In the in vitro transcribed RNA, the transfection itself can play a role in altering the RNA; while for in vitro experiments, we lack the protein interactors naturally present in the cells that might prevent the binding of a molecule that is found in these conditions, or there might be compounds that interact with complexes of RNA and proteins and, therefore, will not be identified in experiments performed in vitro [55].

Moreover, it is important to consider a general limitation of the dual luciferase approach: viral RNA can have, as well, long distance interaction. For instance, the interaction between the 3′ untranslated region (UTR) and the 5′ UTR of different flaviviruses is fundamental for viral replication [56], or distal elements can play a role in the modulation of the PRF [57], and these functions will not be evaluated by cloning only the small RNA portion of interest. This critical aspect was further evidenced by recent research showing that a dual luciferase assay, in which a portion of 88 nucleotides or 2 kb of the −1 PRF of SARS-CoV-2, resulted in a drastically different rate of frameshift (17% vs. 42%) [32].

In the design of the experiment and of the construct, it is important to take into consideration cloning a consensus sequence rather than a sequence of a specific isolate, to assess the effect of the drug on a sequence that is likely to be shared among different circulating strains of the virus of interest.

Considering all these limitations, while recognizing the importance of these high-throughput assays for initial screening, the evaluation of the selected compounds with wild-type viruses is a fundamental step in the evaluation of the identified molecules; however, in numerous works on RNA targeting antivirals, this crucial experiment is missing.

### 2.4. Validation with WT Virus

The evaluation of the antiviral activity of the compound against the wild-type (WT) virus in infected cells is a fundamental condition for finding true viral RNA interactors. The identification of potent inhibitors with other techniques can result in complete failure of activity with a WT virus for two main reasons, either the compound has poor cell permeability or the targeted RNA cannot be bound due to the presence in a discrete zone of the cell not being accessible to the compound. For instance, viral RNA could be associated with nucleoproteins, and this might mask the interaction site with the small molecule.

The other important point to verify with the natural infection is the possible insurgence of resistance. The selection of resistance in vitro could reveal if the site targeted can be easily mutated, therefore hampering the further development of the molecule as an antiviral. However, the eventual selection of resistance is an important verification of the binding site that can be then re-engineered in dual luciferase assays or modeled to understand the change of the binding pocket of the molecule, as was previously done for inhibitors of the RNA of enterovirus 71 [7,13]. Importantly if the resistance mutation leads to the identification or validation of the binding site, it can also open the possibility of modifying the small molecule with a medicinal chemistry approach that could ameliorate the inhibition profile.

## 3. Targetable Viral Elements

### 3.1. Frameshifting Element of SARS-CoV-2

The genome of coronaviruses has two superposed ORFs, named ORF1a and ORF1b. At the end of ORF1a, a stop codon is present; however, upon a −1 PRF, the ribosome can change the reading frame, skip the stop codon, and allow the translation of the full ORF1ab with the expression of the replication machinery encoded in the ORF1b. The −1 PRF is allowed by the presence of a slippery sequence on viral RNA and of a pseudoknot. The secondary structure of the pseudoknot has been investigated with multiple methods [6,23,24,58] and different conformations have been proposed, associated either with free RNA or RNA in proximity to the ribosome. Moreover, some discrepancies are observed between the structure determined by in vitro transcribed RNA and the structures determined by DMS-MaP or SHAPE-MaP [32].

Since the importance of the −1 PRF was known for other coronaviruses, research on this target was already started before the COVID-19 pandemic, especially on SARS-CoV-1. In particular, a compound, 2-{[4-(2-methylthiazol-4-ylmethyl)-[1,4] diazepane-1-carbonyl]-amino}-benzoic acid ethyl ester (MTDB), was identified to be active against −1 PRF sequence of SARS-CoV with a virtual screening [40]. The compound proved then to be active in dual luciferase assays and proposed to act on the plasticity of the pseudoknot [59]. At the beginning of the COVID-19 pandemic, due to the nearly identical sequence of the SARS-CoV-1 and SARS-CoV-2 pseudoknots, the compound MTDB was tested and its activity on the −1 PRF was verified [59]. However, in a different study, the compound did not show activity against the frameshift in infected cells [6].

Several compounds proposed to act on the −1 PRF have been identified through high throughput screening or specific design. In particular, the compound merafloxacin was identified with a high throughput screening: a dual fluorescence assay, based on a plasmid containing mCherry, the sequence of the junction between ORF1a and ORF1b, and a GFP coding sequence in a −1 frame [11]. The activity was verified as well in a traditional dual luciferase assay and with the WT virus with multiple specific assays. The activity was retained against the −1 PRF of other beta-coronaviruses. However, in the presence of several point mutations in the −1 PRF, the compound retained activity, making unclear the exact binding pocket and mechanism. Importantly, the activity of merafloxacin was verified by independent groups [6,12].

Our study, instead, evidenced the activity of one aminoglycoside, geneticin, against the −1 PRF [12]. Aminoglycosides are known to interact with RNA and we tested one of the most permeable in human cells to verify if it was active against SARS-CoV-2. Its activity in the micromolar range is conserved against multiple variants and in human-derived respiratory epithelia. Moreover, we could not select resistance upon multiple passages. Through in silico analysis, we identified the possible binding pocket and designed mutations to occlude it. In the presence of these mutations, in a dual luciferase assay, both geneticin and merafloxacin lost activity, validating our mechanism.

Additional compounds were identified with dual luciferase/fluorescence assays [60,61] (Table 1) and other strategies to target this genomic region have been investigated, for instance, pseudoknot targeting Cas13b [62] or antisense nucleotides [63] to prevent the correct conformation of the pseudoknot.

However, further optimization is required to obtain compounds endowed with stronger antiviral activity and drug-like properties against this target, also considering that the molecules were not tested in vivo. However, these preliminary results evidence that the target is valuable in cells and human-derived tissues without major issues of toxicity. This is possibly due to the presence of few human genes known or predicted to have pseudoknots (for example telomerase RNA, the small subunit of the ribosome, PEG10, and Ma3 [65,66]).

Additionally, the highly conserved structure of the −1 PRF among beta-coronaviruses suggests that optimization of compounds on this target might be useful for newly emerging coronaviruses.

### 3.2. Internal Ribosomal Entry Site

Different viruses rely on secondary or tertiary structures in their 5′ UTR in order to mediate multiple functions for the viral life cycle: the most important, for viruses with uncapped RNA, is the IRES, which allows viral protein translation. IRES are present in the *Picornaviridae* and in the *Hepacivirus*, *Pegivirus*, and *Pestivirus* genera of the *Flaviviridae*. Moreover, they are present in the *Dicistroviridae*, a family of viruses infecting arthropods and insects.

The IRES presents different structures in different viral families [67], and even in between viral families [68]. However, they all mediate two functions: they allow the recruitment of the initiation complex and the remodeling of the ribosome to allow translation. Multiple loops are present to interact with the ribosome, with elongation factors and with accessory proteins that positively regulate the translation. Altering key residues and loops can result in potent inhibition of the viral life cycle.

Few viral RNA inhibitors have been tested in vivo; however, to the best of our knowledge, the majority have as a target the IRES of enterovirus 71 (EV71). The flavonoid prunin was identified with a screening in which the IRES of EV71 was cloned upstream of a luciferase gene. The compound was then further studied and revealed potent antiviral activity from in vitro to in vivo [7]. The mechanism of action has been identified through the selection of resistance: mutations in the stem-loop (SL) two of the IRES were proven to be responsible for the loss of efficacy of the molecule. Moreover, it was shown that the binding of prunin to SL2 resulted in modified binding of one of the IRES-associated proteins, heterogeneous nuclear ribonucleoprotein K (hnRNP). Different flavonoids (kaempferol and apigenin) were also shown to inhibit the IRES of EV71 [69,70], and later to be active in mice [71]. However, the mechanism of action of kaempferol was not deeply investigated while apigenin is proposed to work through the prevention of binding of the viral RNA to hnRNP, similarly to prunin.

Another compound showing activity on EV71 IRES is emetine [72], identified through screening for a cytopathic effect on infected cells, was then shown to influence IRES-dependent reporter expression. Moreover, its activity was conserved as well against other members of the *Picornaviridae* (Table 2) and was also effective in vivo [72]. However, emetine was shown to exert antiviral effects against multiple viruses and, therefore, it is possible to consider it as well an aspecific mechanism [73].

A different compound was found to be effective against the SL2 of EV71 through a different mechanism of action. The compound DMA-135 was identified with a screening with a fluorescent indicator displacement assay and validated through NMR spectroscopy and isothermal titration calorimetry [74]. Its activity was verified in vitro and, through resistance and structural studies, it was shown to work through stabilization of the RNA with a host protein (AUF1) resulting in the inhibition of viral translation.

Another interesting example of IRES targeting compound is amantadine. This compound is well known to be an inhibitor of the ion channel of the Influenza A virus, necessary for viral entry. However, several reports showed its activity on IRES-mediated translation of different members of *Picornaviridae* (EV71, Cardiovirus A, and Hepatitis A virus (HAV)) [75,76] and debated results on Hepatitis C virus (HCV). Different groups found activity on members of the *Picornaviridae*, with dual luciferase assays in which the compound inhibited only the IRES-mediated translation of HAV, EV71, and Cardiovirus A [75]; subsequently, however, the compound showed activity as well in cells infected with HAV [76]. Amantadine even showed some activity in a clinical trial for HCV-infected patients not responding to interferon [77]. However, successive analysis showed that the effect seemed not to be virus specific, nor IRES mediated [78].

Nevertheless, the IRES of HCV was successfully targeted by a series of different compounds [79,80,81,82] mainly binding to the SL2a loop. Outside of the discussed members of *Picornaviridae*, or the ones shown in Table 2 and HCV, however, not many targets were explored.

**Table 2 ijms-24-13500-t002:** Molecules active on viral IRES.

Target	Molecule	Identification	Activity
IRES (SL2) of EV71, CVA6, CVA7, EchoV7, CVB5	Prunin [7]	Dual luciferase assay	Cells and mice with WT virus
IRES (SL2) of EV71	DMA-135 [74]	Fluorescent indicator displacement assay	Cells with WT virus
IRES of EV71	Kaempferol [69]	Dual luciferase assay	Cells with WT virus, mice [71]
IRES of EV71, FMDV	Apigenin [83,84]	Targeted testing	Cells with WT virus, mice [71]
IRES of EV71	Idarubicin [85]	Screening with WT virus	Cells with WT virus
IRES of EV71, EVD68, Echov6, CVA16, CVB1	Emetin [72]	Screening with WT virus	Cells with WT virus, mice
IRES of EV71, ECMV, HAV	Amantadine [75,76]	Dual luciferase assay, targeted testing	Cells with WT virus
IRES of HCV (SL2a)	Benzimidazole derivatives [81,82]	Mass spectrometry on RNA model and structure–activity relationship	Screening with mass spectrometryActivity on replicon of HCV in cells
IRES of HCV (SL2a)	DAP compounds [80]	Targeted design	Luciferase assay with IRES
IRES of HCV	Geneticin [79]	Targeted testing	Cells with WT virus
IRES (SL3a) of FMDV	IRAB [86]	Targeted design	Cells with WT virus

In the *Picornaviridae* family, it will be important to continue the research of small molecules active on the IRES of other members, for instance, enterovirus D68 for which mutations in the IRES have been linked to increased pathogenicity [87] or for widely circulating pathogens, such as rhinoviruses, for which nowadays we do not have therapeutic options. Even dicistroviruses might be a potential target for IRES inhibitors since they have been proposed to potentially infect humans [88].

In addition to the known viruses bearing IRES, additional viruses have been proposed to have IRES, for instance, hepatitis E virus to express the ORF4 [89] or flaviviruses to initiate the translation with a cap-independent mechanism [83]. It is, therefore, expected that the list of compounds targeting IRES and of targeted viruses will grow in the next years.

In parallel with the research on new targets, a focus must be as well the research of new molecules specifically acting on viral IRES in order not to risk targeting human IRES, which are involved in the translation of genes involved in stress and apoptosis [90], processes often ongoing in infected cells. Despite part of the compounds listed in Table 2 showing some aspecific activity, the results of prunin [7] and DMA-135 [13,74] are encouraging since a specific viral RNA targeted mechanism of action was identified thanks to the selection of resistance. These results prove that achieving specific viral inhibition by targeting the viral IRES is possible and the activity can be maintained up to in vivo [7].

### 3.3. RNA Elements of HIV

A largely studied field of viral RNA targeting structures has been HIV. Several RNA structures were identified as targets for antivirals: frameshift site Gag:Pol, TAR, RRE, and Psi Stem Loop 3.

Gag is a polyprotein encoding for the matrix, capsid, nucleoprotein, and p6. After expression, it is involved in the assembly, release, and maturation of HIV particles. However, a polyprotein named GagPol containing the domains of Gag and three additional ones (protease, reverse transcriptase, and integrase) can be translated after a −1 PRF [91]. The frameshift regulates the stoichiometry of the expression of Gag and GagPol proteins and its alteration results in a loss of viral fitness representing an interesting antiviral target [91]. In opposition to what was seen for SARS-CoV-2, inhibitors of HIV were shown to increase the frameshifting activity. The majority of compounds active on this target derive from the hit of an initial screening [92], followed by several attempts of chemical optimization of the molecule [92,93,94,95] (Table 3), which resulted in compounds showing micromolar activity against different HIV strains [95].

To replicate efficiently, HIV uses TAT protein. TAT recognizes the TAR element on the nascent 5′ end viral RNA and recruits host factors to enhance transcription [96,97]. Modifications of this conserved TAR element impact negatively its interaction with TAT, affecting the replication of HIV [96]. Therefore, TAR-targeting molecules represent an attractive antiviral target, as demonstrated by the large number of small molecules identified (Table 3). Among them, netilmicin was identified with a path like the one described in this review, starting from the determination of the RNA structure, to in silico screening, and then verification of the mechanism of action and the specificity [10]. A library of about 51,000 molecules was screened in silico against 20 conformations of TAR based on molecular dynamics and NMR to obtain 57 hits. The selected hits were then narrowed through fluorescent assays to assess binding to TAR and inhibition of TAT–TAR interaction. The antiviral activity of netilmicin was further verified with WT HIV. Additionally, netilmicin was shown to be specific by interacting with the different substructures of TAR without being affected by the presence of tRNA [10].

RRE, an RNA structure containing multiple bulges, internal loops, and hairpins, is present on unspliced and partially spliced viral RNA [98,99]. In the nucleus, Rev, an accessory viral protein, will interact with RRE. This complex, together with host proteins, can be then exported outside the nucleus where the viral RNA is released to be packed in the new HIV virions or transcribed for the expression of viral proteins [99,100]. Affecting the interaction of Rev with RRE impacts HIV infection [100], as shown by the different molecules identified (Table 3). The most studied antiviral against RRE is an aminoglycoside, neomycin B [101,102,103,104,105,106]. Although detected in 1993 to inhibit Rev–RRE interaction [106], and then shown to bind RRE [102,103], neomycin B showed only limited antiviral activity against the WT virus [101].

During viral assembly, Gag interacts with viral RNA and, in particular, with the Psi packaging domain located in the 5′UTR of HIV [107]. Destabilizing this interaction impacts the release of HIV virions containing new viral RNA. One small molecule, named NSC260594, binds to the Psi SL3 structure, decreasing its flexibility. Gag interaction is, therefore, reduced with a decrease of viral RNA released extracellularly [107,108].

Most of the structures targeted on HIV elements are hairpins highlighting that targeting simpler structures of RNA can also be successful. Unfortunately, to our knowledge, none was studied in vivo. This limitation is possibly due to the presence of many effective conventional antivirals, or to the availability of small animal models.

**Table 3 ijms-24-13500-t003:** Molecules active on HIV RNA elements.

Target	Molecule	Identification	Activity
Frameshift site	RG501/DB213 [109,110]	Split luciferase assay	Cells with WT virus
Frameshift site	Compounds 4 and 5 [92], N-methylated derivatives [94], triazole derivatives [95], and lower size derivatives [93]	Surface plasmon resonance on biotinylated RNA and chemical optimization	Cells with pseudotyped virus [92,94] and cells with WT virus [95]
TAR	T0516-4834 [111]	In silico screening	Cells with pseudotyped virus
TAR	Compound 4 [112]	Screening with targeted RNA	Cells with WT virus
TAR	6-Aminoquinolones [113] and derivatives [114,115]	Targeted testing	Cells with WT virus
TAR	Netilmicin [10]	In silico screening	Cells with WT virus
TAR	Acetylpromazine and Prochlorperazine [46,116]	In silico screening	Binding assay, Cells with coexpression of TAT/TAR [46]
TAR	Furimidazoline/DB60 [117]	Targeted testing	Chronically infected cells
TAR	Nucleobase-amino acid conjugates [118]	Targeted design	Cells with WT virus
TAR	Compound 17 and 20 [119]	Targeted design	Cells with WT virus
TAR	460-G06 and 463-H08 [120]	Fluorescence resonance energy transfer	Cells with WT virus
TAR	Amiloride derivatives [121,122]	Targeted testing and chemical optimization	Binding assay
TAR	Purine substituted [123]	Targeted design	Cells with coexpression of TAT–TAR and with Simian Immunodeficiency Virus
TAR	Aminoglycoside-arginine conjugates [124,125]	Targeted design	Binding assay and cells with Equine Infectious Anaemia Virus
TAR	Compound 3ba and 3ca [126]	In situ cycloaddition	Fluorescence resonance energy transfer-based displacement assay
RRE	Neomycin B [101,102,103,104,106] and conjugates [127], Neamine [104], and dimers [128]	Targeted testing, fluorescence anisotropy [104]	Cells chronically infected, [106] cells with WT virus [101,104]
RRE	P-terphenyls substituted [129]	Targeted design	Cells with WT virus
RRE	Mitoxantrone, clomiphene, ciprofloxacin and cyproheptadine [104], Compound 1a (benfluron), 1b, and 2a [130]	Fluorescence anisotropy	Cells with WT virus
RRE	DB340, DB182, A132/DB247 [105,131,132]	Gel band shift assay	Binding assay
Psi stem loop 3	NSC260594 [107,108]	Fluorogenic destabilization assay	Cells with pseudotyped and WT virus

## 4. Other Targets and Conclusions

The targets described in this review are the ones for which more literature is available; however, other viral RNA targets were investigated such as the 5′ and 3′ conserved nucleotides on the segmented genome of influenza [133] or the frameshift of JEV [134] or the 5′UTR of coronaviruses [135]. However, several works have revealed structures or predicted conformations that have not been investigated yet in terms of inhibition. Moreover, new targets are being discovered thanks to a better understanding of the structure of viral RNA in living cells, and thanks to the techniques reviewed here, the identification and evaluation of small molecules is feasible. Therefore, future research in this field will be thrilling, on one side for new potential antiviral development but also with a complementary approach to use small molecules to reveal new functions of viral RNA and identify new interactors.

## Figures and Tables

**Table 1 ijms-24-13500-t001:** Molecules active on the −1 PRF of SARS-CoV-2.

Molecule	Identification	Activity
MTDB [40,59,64]	In silico screening	Dual luciferase assay with −1 PRF sequence
Merafloxacin [6,11,12]	Dual fluorescence screening	Cells with WT virus
Geneticin [12]	Targeted testing	Cells and human-derived tissues with WT virus
Aminoquinazoline derivatives [60]	Array screening	Dual luciferase/fluorescence assay with −1 PRF sequence
C5 and C5-ribotac [61]	RNA-binding assay	Dual luciferase assay with −1 PRF sequence

## Data Availability

Not applicable.

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
