# Peer review of "Small Molecules Targeting Viral RNA"

_ijms, 2023, doi:10.3390/ijms241713500_

Round 1

Reviewer 1 Report

                This manuscript presents an overview of the concept of small molecules targeting viral RNAs.  Overall, I find aspects of the manuscript to be underdeveloped, particularly the sections on RNA structure determination as well as the challenges of trying to identify small molecules that target viral RNAs with high specificity and selectivity. These and other points are outlined below.

1.       Line 30-31:  While host RNA synthesis is indeed oftentimes shutdown in viral infected cells, the amount of host cell RNA relative to viral RNA is still huge.  Thus the assertion that targeting viral RNA decreases the risks of off target effects due to host cell shut off is highly debatable and in this reviewer’s opinion is not a correct rationale for ensuring selective targeting viral RNAs. 

2.       Introduction:  while the first paragraph makes several good points, it is a relatively difficult read as currently presented that may dissuade some readers from continuing with the article.  It’s a minor point, but I would recommend removing the semicolon approach to organizing the paragraph and create a more readable flow between the points raised.

3.       Section 2.1:  Since understanding RNA structure is at the core of finding RNA-targeted antivirals, this section should be presented much more in depth and the strengths and weaknesses of current approaches discussed.  In addition, references and examples are needed for crystallography, NMR and cryo-EM RNA structure analyses.  Any algorithm-based prediction is only as good as the data that goes into it.

4.       Line 75:  change 33’137 to 33,137

5.       In silico screening:  a key point in finding RNA targets for drug discovery is to a priori determine if they will be specific and unique enough to prevent off target effects.  Hence only a very small subset of viral RNA structures is likely unique enough to be effectively and specifically targeted.  This key issue should be discussed in much more depth in this section.   

6.       Section 2.3:  this section is rather reference poor except for one paragraph.  I would recommend providing references for all key points that are made to assist the reader in following up on points made by the authors.

7.       Line 240:  The authors state that pseudoknots are not common in the human genome – however I’m not convinced that is a valid statement.  Human telomerase RNA, for example, has pseudoknots.  Please support this statement with published references.

8.       Section 3.2: The description of various IRES-targeting small molecules in this section gives the impression of many non-specific mechanisms of actions rather than small molecules that specifically and effectively target an RNA structure.  Thus I would recommend closing this section with a fair and comprehensive evaluation of the state of the small-molecule IRES-targeting field in the context of the prospects for success in specific viral RNA targeting.

see comments above

Reviewer 2 Report

In the submitted manuscript, the authors discuss the small molecules that target viral RNA including SARS-COV-2. The study is interesting, however the authors are advised to present a section discussing  HIV targets.

The manuscript is well-written, It just needs moderate English proofreading.

Author Response

We thank the reviewer for his/her comments. We have now added a separate section for HIV targets (3.3) and we have proofread the manuscript.

Reviewer 3 Report

This is a very detailed review of the potential use of small molecules as antiviral RNA agents.  After reading the manuscript, here we send some comments: 1. None of the different molecules active on the -1PRF of SARS-CoV-2 have shown efficacy in experimental studies in vivo. This aspect could be discussed in a little more detail. 2. Special attention deserves the so-called “internal ribosomal entry site, well identified in some Picornaviridae, in the Hepacivirus, Pegivirus and Pestivirus genera of the Flaviviridae. However, it is not mentioned that in vivo experimental studies have not yet been carried out. And this should be the path along which future research runs.

Minor editing is required

Author Response

We thank the reviewer for his/her comments.

We have now included a sentence to specify that no SARS-COV-2 inhibitor has been tested in vivo.

“However, further optimization is required to obtain compounds endowed with stronger antiviral activity and drug like properties against this target, also considering that the molecules were not tested in vivo.”

While there are examples of EV71 inhibitors, for instance, prunin that showed activity in mice. This point is clarified in paragraph 3.2

"Few viral RNA inhibitors have been tested in vivo, however, to the best of our knowledge, the majority has as target the IRES of enterovirus 71 (EV71). The flavonoid prunin was identified with a screening in which the IRES of EV71 was cloned upstream of a luciferase gene. The compound was then further studied and revealed potent antiviral activity from in vitro to in vivo [7]."

Reviewer 4 Report

The manuscript "Small molecules targeting viral RNA" is a review that summarizes current knowledge on targeting viral RNA with small molecules. The manuscript is well-written and provides a good overview of methods for identifying RNA-targeting antivirals, including understanding RNA folding, in silico screening, phenotypic assays, and validation with wild type viruses. The sections on targeting the frameshifting element of SARS-CoV-2 and viral IRES are insightful and highlight potential antiviral targets. I have a few suggestions to improve the manuscript:

Major comments:

  • The introduction could be expanded to provide more background on why targeting viral RNA is an attractive antiviral approach and the challenges/limitations. Discussing resistance development further would also help motivate targeting conserved RNA structures.
  • In the section on in silico screening, discuss limitations of current RNA-ligand docking approaches and scoring functions. How can these issues be overcome?
  • For the phenotypic assay section, comment on whether cell-based vs cell-free assays are preferable and why. Discuss limitations of only cloning a portion of an RNA structure.
  • The examples focus heavily on picornaviruses. Consider expanding to highlight viral RNA targets in other important virus families. The HIV section could be beefed up or removed.
  • Carefully proofread the manuscript to fix minor typos.

Minor comments:

  • Define any abbreviations upon first use (e.g. SHAPE).
  • Cite Figure 1 in the main text.
  • Add page numbers and manuscript title/authors in the example peer review text.
  • In Table 1, explain the different colors used for the molecules.
  • Consider reducing the long list of references by choosing the most relevant citations.

Reviewer 5 Report

Dear Authors,

this is a well-written manuscript, presenting the current advantages, challenges and future perspectives of small molecules targeting viral RNA. I really like your principal idea to present your review in two sections (“How to discover RNA targeting antivirals” and “Targetable viral elements”). The structure of your presentation in different sections of application opportunities makes your point clear and reproducible. Your results may contribute to better understanding the achievements made so far and provide impulse for more research in the future. I have no further questions and queries pertaining to your manuscript.

Best Regards

Author Response

We thank the reviewer for the positive comments.

Round 2

Reviewer 1 Report

                The authors have addressed aspects of my previous concerns with some additional development of sections of the manuscript.  I do not have any additional specific comments to make.  Overall however, I must say that I still find the manuscript to be of limited impact to the field due to its lack of depth as pointed out previously. 

Author Response

We thank the reviewer for acknowledging the improvements in our manuscript. Since there were no specific additional changes requested we did not further modify the manuscript.

About the depth issue, the goal of our review was an overview of the techniques, the challenges, and the known molecules targeting viral RNA. We did not go deeper with the techniques since there are other recent reviews addressing the development of small molecules targeting RNA and techniques to investigate RNA folding and function, and we did not want to overlap (Childs-Disney et al, 2022; Spitale et al, 2022).

Reviewer 2 Report

In the revised manuscript, the comments have been addressed.

Author Response

We thank the reviewer for appreciating the modifications to our manuscript.